# Epidemiologic and Clinic Characteristics of the First Wave of the COVID-19 Pandemic in Hospitalized Patients from Galați County

**DOI:** 10.3390/jcm10184210

**Published:** 2021-09-17

**Authors:** Mihaela-Camelia Vasile, Anca-Adriana Arbune, Gabriela Lupasteanu, Constantin-Marinel Vlase, George-Cosmin Popovici, Manuela Arbune

**Affiliations:** 1The School for Doctoral Studies in Biomedical Sciences, “Dunărea de Jos” University of Galati, 800008 Galati, Romania; mihaela.vasile@ugal.ro (M.-C.V.); constantin.vlase@yahoo.com (C.-M.V.); popovicigeorge1989@gmail.com (G.-C.P.); manuela.arbune@ugal.ro (M.A.); 2Neurology Clinic, Fundeni Clinical Institute, 022328 Bucharest, Romania; 3The School for Doctoral Studies in Medicine, “Ovidius” University of Constanta, 900527 Constanta, Romania

**Keywords:** COVID-19, clinical features, case fatality rate, epidemiology, Romania

## Abstract

The first cases of COVID-19 were reported in Wuhan Province, in China, in December 2019, spreading rapidly around the world. The World Health Organization (WHO) declared this pandemic at the beginning of March 2020 and, at the same time, the first patient in Galați County was confirmed. Both the global and the regional epidemiological evolutions have taken place with variations in incidence, which have been graphically recorded in several “waves”. We conducted a retrospective study on cases of COVID-19 infection, hospitalized between March and June 2020 in an infectious diseases clinic from Galati, in the south-east of Romania. The present paper describes the “first-wave” regional epidemiological and clinical-biological features and the evolution of the COVID-19 pandemic. A poor outcome was related to late presentation to hospital, old age, and over six comorbid conditions including Alzheimer’s disease. The high death rate among people from long-term care institutions is the consequence of the cumulative risk factors associated with immune senescence and inflammation, while COVID-19 is more likely a contributing factor to lower life expectancy.

## 1. Introduction

The first cases of COVID-19 were reported in Wuhan Province, in China, in December 2019, spreading rapidly around the world. The WHO declared this pandemic at the beginning of March 2020, while in the WHO European Region, the first case was reported from France on 24 January 2020, with the individual having travel history to China. Very soon, by 11 March, the number of cases in the European Union and United Kingdom had increased to 17,413 [1,2]. Romania reported the first cases at the western border at the end of February. Both the global and regional epidemiological evolutions have so far taken place with variations in incidence, recorded in epidemic “waves”.

In Galați County, located in the south-eastern region of Romania, the “first wave” began on 7 March and lasted until the end of June. The first COVID-19 case was a man who had travelled to Italy. In the next 2 weeks, import cases were predominantly diagnosed, with two main outbreaks being identified in seniors who were on vacation in Turkey, as well as family outbreaks or isolated cases after trips to England, Italy, France, Israel, and Egypt. At the beginning of April, cases of COVID-19 appeared in two senior care centers, along with the increase in the number of cases among medical staff in hospitals and other care units, as well as within the community.

COVID-19 outbreaks in the “first wave” have also been reported in a high number of long-term care facilities across Europe, from France, Belgium, Germany, Norway, Sweeden, Spain, Ireland, Scotland, and the United Kingdom, with high rates of morbidity and case fatality in residents and high rates of staff absenteeism [3]. However, the characteristics of initial COVID-19 surveillance in Europe suggested underreported cases from long-term care facilities, owing to asymptomatic cases, transmission dynamics, and low availability of testing [4].

The aim of this study is to describe the clinical features, biological effects, and outcomes of patients with COVID-19 during the “first wave” in the south-eastern region of Romania.

## 2. Materials and Methods

The study was performed retrospectively on cases of COVID-19, hospitalized between March and June 2020, in one of the departments from the Infectious Disease Clinical Hospital “Sf. Cuv. Parascheva” Galați. All cases were diagnosed according to the case definitions valid at that time, including a sudden onset of cough, fever, or shortness of breath with no other aetiology that explains the clinical presentation, and confirmation by RT-PCR-SARS-COV2 of a sample from a nasopharyngeal swab [5]. We collected the general clinical data recorded for continuous hospitalization, according to the forms recommended by the WHO for COVID-19 case registration in clinical trials [6]. Duplicated cases, readmitted in hospital during the same episode of illness, were excluded. The risk of death after 1 year was estimated with the Charlson prognosis index, which evaluates 13 co-morbidities, each weighted with a score from 1 to 6, added up in a final score for each patient. The score is 0 if no co-morbidities are found and has a maximum of 29 with 100% estimated risk of mortality [7]. The mortality at 30 days after discharge was monitored according to the national population statistics register and was correlated with the Charlson index. Aging people over 61 years old were separated into two groups, according to the status whether or not they lived in a long-term facility care. According to the legal and methodological regulations of the Romanian National Centre for Infectious Diseases, during the analyzed period, all confirmed cases were hospitalized, with criteria for discharge at home being two consecutive negative RT-PCR-SARS-CoV-2 virological tests from nasopharyngeal swabs [8].

We statistically analyzed the demographic, clinical, and laboratory records data collected according the local protocol for the management of COVID-19 infection. The case fatality rate was considered as the ratio between the number of deaths divided by the number hospitalized cases of COVID-19 during the same time, and then multiplied by 100. Surveilling the deaths occurring in the 30 days following hospital admission, we have reported 30-day mortality. Data were collected in Office Excel 2010 and the statistical analysis used the software program SPSS version (24.0), with descriptive statistics components and correlations, taking into consideration a limit of statistical significance of *p* < 0.001. For continuous variables, we calculated the mean, standard deviation (SD), or median, evaluating the differences using the two-sample *t*-test or the Mann–Whitney-U test. The categorical variables were expressed as a percentage (%) and were compared using Pearson χ^2^ tests (for 2 × 2 tables) or Fisher exact tests (for tables larger than 2 × 2).

## 3. Results

### 3.1. Demographic Characteristics

There were 115 patients confirmed to have COVID-19, ranging from new-born babies to people 96 years of age, with a mean of 60.7 ± 24.5 years and a median of 67 years of age.

Men were prevalent, with a sex ratio of 1.5. Most patients lived in urban areas (96.4%). From the point of view of occupation, over half of the patients were retired, and 12.5% of the active people had a medical profession. The most common chronic co-morbidities were hypertension (43.5%), chronic heart disease (23.5%), chronic neurological diseases (36.5%), obesity (26.5%), diabetes (13.9%), and chronic kidney disease (10.4%) (Table 1).

The Charlson co-morbidity index ranged from 0 to 10, with a mean of 4.36 ± 2.98 and a median of 5, signifying a 6.38 relative risk of death [7].

### 3.2. Clinical Features

The mean duration of symptoms from onset to hospitalization was 3.1 ± 3.2 days. The symptoms appeared after hospitalization in three cases with virological diagnosis after the epidemiological investigation. The main reasons for testing/hospitalization were as follows: 62.61% (72) fever above 38 °C, 58.52% (65) cough, and 43.48% (50) breathing difficulties such as shortness of breath or a compelling feeling of shortness of breath, bringing respiratory symptoms in the foreground.

The clinical parameters observed at hospitalization mentioned the alteration of the following vital parameters: 6% fever > 38 °C, 53% tachycardia > 90/min, 41.7% tachypnea > 20/min, 19.1% SO_2_ hypoxemia < 93%, and 2.6% Glasgow score < 15 (Table A1 in Appendix A).

At hospitalization, particular situations were also mentioned, such as altered consciousness (three cases) and dehydration (four cases).

In our study, the clinical manifestations mentioned during hospitalization had variable frequencies. The most common symptom was fatigue, at 34.78% (40/75). Between 10% and 20% of patients reported cough (22/93), wheezing (21/94), diarrhea (20/95), dysphagia (19/96), loss of smell (18/97), loss of taste (14/101), and myalgias (13/102). Less than 10% of the cases reported abdominal pain (11/104), vomiting/nausea (9/106), chest pain (8/107), headache (8/107), arthralgia (5/110), nasal obstruction (3/112), and rash (3/112) (Table 2).

### 3.3. Laboratory Characteristics

The biological picture ranged from normal values to changes interpreted in the context of pre-existing co-morbidities and complications of COVID-19 infection. There were also some associated hematological changes, such as the following: 20% anemia (Hb < 11 g/dL), 16.5% leukopenia (<4000/mm^3^) or leukocytosis (>12,000/mm^3^), and 16.5% thrombocytopenia (<150,000/mm^3^) (Table A2).

The most common biochemical changes in blood were the increase in D-dimers (71.6%), C-reactive protein (60%), lactic acid (54.9%), and ferritin (37%). Cytolysis markers were increased in one-third of patients (Table A2). These results reflect the pathogenic changes specific to the SARS-Cov2 infection, inflammation, hypercoagulation, cytolysis, and tissue hypoxia. Procalcitonin displayed significantly elevated values in four cases.

### 3.4. Radiological Data

The radiological examination was available from variable stages of infection. Unilateral or bilateral images specific to COVID-19 infection were revealed in 64.35% of cases, mostly those described as condensation opacities in the airspace or as “ground glass” images, especially in the peripheral lung areas and in the lower lobes. Pleural reactions were described in three cases and pneumothorax complicated the evolution of the disease in two cases.

### 3.5. Outcomes of COVID-19

The complications associated with lung damage were the following: acute respiratory distress in 55.6% of cases (64), sepsis in 2.6% (3), and coagulation disorders (17). During hospitalization, in 14.7% of patients, *Clostridium difficile*-associated infections occurred. *Pseudomonas aeruginosa* (2), *Klebsiella pneumoniae* (1), *Proteus* spp. (1), and methicillin sensible *Staphylococcus aureus* (1) have also been reported.

According to the severity classification of cases, we reported 22.6% (26) critical forms, 24.35% (28) severe forms, 45.22% (52) medium forms, and 7.82% (9) mild forms.

The outcome was favorable for 73% of patients, of which 64.3% (74) were cured when discharged and 8.6% (10) were in an improved state of health and were transferred to other hospital units named “COVID Support”. One patient, who was clinically stable, was transferred to a thoracic surgery unit owing to a radiological suspicion of pneumothorax.

An unfavorable outcome was appraised for 27% of the patients, considering that 12.1% (14) patients died in hospital and 13.9% (16) worsened and were transferred to intensive care units. The cumulated case fatality rate at 30 days after admission was 28.69% (33/115), which better highlighted the impact of COVID-19 on mortality. The 30-day fatality rate was correlated with a Charlson score (adjusted for age) over 6, the association of Alzheimer’s disease, and the duration of symptoms before hospitalization over 4 days (Table A3).

The hematological, biochemical, and clinical symptomatology parameters or the sex of the patients were not significantly correlated with death.

Most deaths were reported in people over 60 years old, with a 30-day case fatality rate of 43.3% (29/67), mainly involving residents from long-term care facilities. Compared with the same age group of home care patients, the residents from long-term care facilities are 11 years older and accumulate more co-morbid conditions according to the Charlson score, including the majority of Alzheimer’s disease cases of the study group (32/31).

## 4. Discussion

From the beginning of March to the end of June 2020, the Regional Public Health Authority of Galați County reported 912 cases of COVID-19, out of which 11.5% were evaluated in our study. The small number of children among the whole group could be explained by the low addressability of pediatric patients to the Infectious Disease Hospital, by the lower rate of infection, and by the predominance COVID-19 asymptomatic or mild infections during childhood. Furthermore, 59.13% of patients were older than 60 and 29.5% were even over 80, explaining the increased frequency of chronic co-morbidities and the significant proportion of high-risk patients, according to the Charlson score. Alzheimer’s disease and other dementias accounted for 73.8% of neurological diseases and 27% of all hospitalized cases of COVID-19. A high impact of Alzheimer’s disease on severe outcome and death was found in our study. This is concordant with reports from other European countries, regarding the increased risk of dementia for the severe outcome and death of patients with COVID-19 [7,9].

Risk factors for dementia are lifestyle, environment, and genetic background. Several studies evidenced that the ApoEe4 genotype is associated with dementia and increases the risk of Alzheimer’s disease by 14 times, as compared with the common e3e3 genotype. The ApoE e4 genotype affects lipoprotein function and the associated cardiometabolic diseases, and modulates the phenotype of pro/anti-inflammatory macrophages and the ACE2 receptor expression in type II alveolar pneumocytes. However, the mechanisms that could explain the link between Apo e genotype and COVID-19 severity have not yet been clarified [10,11].

The role of co-morbidities in the severity of evolution and the mortality caused by COVID-19 infection has been evaluated by numerous studies. The most common identified co-morbidities were hypertension, obesity, and diabetes [12], also reported in our study, characterized by the increased prevalence of chronic neurological diseases. The increased level of D-dimers, characteristic of the biochemical profile of the patients in our study, confirms the reports of other studies [13], explaining the increased incidence of thrombotic complications in patients with COVID-19 [14], which potentiates the pre-existing risk of thrombotic events associated with hypertension and diabetes [15].

The medical literature reports various common symptoms related to COVID-19 infection, although no specific symptom has been validated up to the present (Table 2) [16,17,18,19]. In addition, people over 65 years of age or people with associated co-morbidities may have atypical symptoms. In the patients from our study, the rate of anosmia and/or dysgeusia was lower, probably in the context of frequent chronic neurological comorbidities, owing to the neurocognitive inability to perceive or to express these symptoms. Moreover, elderly patients could have anosmia before COVID-19 infection, related to degenerative conditions such as Parkinson’s disease, sequelae of strokes, or dementia. These symptoms may also be underestimated by incomplete medical history or errors in recording the medical data.

Computed tomography examination is recommended as a standard procedure for the diagnosis of COVID-19 pneumonia, but it was not available in our hospital. Underestimation of the pulmonary radiological lesions could be possible, because X-ray imaging is less accurate than CT. The risk of failing the radiologic lesions of the medium cases or at the beginning of the disease is estimated to 18% [20]. The presence of pulmonary radiological changes was reported in 69% of the hospitalized patients and the lesions expanded to 80% in the next 10–12 days [21].

The treatment guidelines for COVID-19 infection had several versions depending on the necessary updates for rapid medical progress, with critical cases referred for specific care in intensive care units.

Mortality data are variable and should be interpreted with caution, depending on population characteristics, hospitalization criteria, and duration of monitoring. A study on the evolution of hospitalized patients in Germany indicated an overall mortality of 22%, with variations between 16 and 53% between unventilated and mechanically ventilated patients, as well as between age groups, from 5% between 18 and 59 years and 72% over 80 [22]. Another study, conducted in a hospital in Milan, at the beginning of the pandemic, showed a mortality of 20.6%, with an average age of 61 [23]. In our study, more than half of the patients are elderly and have an increased risk of co-morbidities. The mortality rate, extended at 30 days after admission, was 28.69%, much higher than 17%, reported in a similar study in Sweden, which monitored 117 patients, mostly elderly [24].

The risk of long-term facility care on COVID-19 morbidity and mortality in our setting, as well as in many other European countries, is controversial. Obviously, the morbidity is related to the high transmission rate of SARS-Cov2, the dependence of caring human contact for old people, and the low adherence to preventive measures of patients with neurocognitive disorders. The relation between a high case fatality rate and living in long-term care facilities is the consequence of interference with older age and a greater number of comorbidities, including Alzheimer’s disease and other types of dementia. The excess mortality in aging people that was reported across Europe during the first wave of COVID-19 was explained as the consequence of the cumulative risk factors associated with immunesenescence and inflammation, while interaction with COVID-19 lowered the life expectancy [25,26].

## 5. Conclusions

The first “wave” of the COVID-19 pandemic in Galați mainly affected the institutionalized elderly with multiple co-morbidities. The main clinical manifestations were fever, cough, respiratory dysfunction, fatigue, and diarrhea, with severe evolution and a high fatality rate. Late hospital presentation and high values of Charlson co-morbidities including Alzheimer’s disease association provide a prediction of the risk of death, being useful for clinical protocols to orientate patients early on towards the most appropriate health care services.

## Figures and Tables

**Table 1 jcm-10-04210-t001:** Demographic characteristics of hospitalized patients with COVID-19.

		*n*	%
Age	≤20 years	11	9.6%
	20–39 years	10	8.7%
	40–59 years	26	22.6%
	≥60 years	68	59.1%
	≥80 years	34	29.6%
Gender	Women	46	40%
	Men	69	60%
Environment	Urban	112	96.4%
	Rural	3	2.6%
Occupation	Employed	48	41.7%
	Medical personnel	6	12.5%
	Non-medical profession	42	87.5%
	Pensioners	67	58.3%
Co-morbidities	Obesity	31	26.5%
	HTA	50	43.9%
	Chronic neurological disease	42	36.5%
	Chronic heart disease	28	23.5%
	Diabetes	16	13.9%
	Chronic kidney disease	12	10.4%
	Chronic liver disease	9	7.8%
	Chronic lung disease	7	6.1%
	Malignancies	5	4.3%

**Table 2 jcm-10-04210-t002:** Comparative frequency of clinical symptoms related to COVID-19 infection.

Symptoms	Frequency				
	Cov-GALAȚI	Cov-Comparative Studies			
	*n* = 115	*n* = 1420	*n* = 24,410	*n* = 370,000	*n* = 11,950
Fever	62.6%	45.4%	78%	43%	77%
Cough	58.5%	63.2%	83%	50%	60%
Dyspnea/shortness of breath	43.48%		17%	29%	19%
Dysphagia	16.52%	52.9%	12%	20%	11%
Rhinorrhea	2.60%	8%	5%		0.5%
Chest pain	6.95%		7%		14.9%
Myalgia	11.3%	62.5%	17%	36%	16%
Arthralgia	4.34%		11%		
Headache	6.95%	70.3%	13%	34%	10%
Hyposmia	15.65%	70.2%	25%	10%	
Hypogeusia	12.17%	54.2%	4%		
Abdominal pain	9.56%		4%		
Vomiting/nausea	7.82%		10%	12%	0.7%
Diarrhea	17.39%		10%	19%	0.9%
Rash	2.60%		0		
Fatigue	34.78%	63.3%	31%	44–70%	38%
Remarks	All forms	Mild/medium forms [9]	All forms [10]	CDC Guide—All forms [11]	All forms [12]

## Data Availability

Not applicable.

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
