# Peer review of "Epidemiologic and Clinic Characteristics of the First Wave of the COVID-19 Pandemic in Hospitalized Patients from Galați County"

_jcm, 2021, doi:10.3390/jcm10184210_

Round 1
Reviewer 1 Report
Introduction
- The authors should consider a less historical introduction. It would be interesting to lay the ground for what the methods and results of the paper are. For example, include more epidemiological information about the pandemic in other countries maybe in Europe. Include data on severity, clinical characteristics and how the pandemic has behaved in other places. It is important to know how transmission started in Galati county, but this could be condensed. Is the Galati county one of the biggest in Romania?
Methods
- Authors should consider giving a better structure to the methods section. Include sections of the methods (i.e. study population, study design, statistical analysis, testing criteria, etc…). Elaborate as much as possible in a concise manner each of these sections.
Results
- Authors should consider narrowing the results section. Including data on therapy and how individuals were treated in this particular group of people can be described in separate analysis.
Author Response
Dear Reviewer,
Thank you for your analysis and the recommendation for improving the manuscript.
Answers to comments:
Introduction
- We have shortage the historical data in introduction and have added more epidemiological information about the pandemic in Europe. Galati is an important region in south-east of Romania, and a harbour city of the Danube River. On our knowledge, our department have managed the firs elderly people infected in long-term care facilities from Romania. This group of patients is consisting of over a half from the hospitalized cases in the first weeks of epidemic.
Methods
- We have revised the methods section.
Results
- We revised the result section, ignoring data on therapy. We’ll consider the suggestion to analyse separate the treatment of this patients.
We’ll improve English language and style asking special counselling from the Editor.
Kind regards,
Manuela Arbune
10 September 2021

Reviewer 2 Report
This paper documents the epidemiological and clinical characteristics of patients with Covid-19 infection from an infectious diseases clinic in the south-east of Romania, during the first wave of infection between March and June 2020. The study was performed retrospectively. In addition to document the initial pattern of infection relatively early in the pandemic, the study also documents the experience in a relatively health resource poor area within the EU. (For example, the hospitals involved in the study did not have access to high resolution CT scans of the chest).
The clinical features including laboratory and radiological findings were similar to those described elsewhere. The rate of anosmia and/or dysgeusia were somewhat lower in this cohort but is likely to be due to influence by the age structure of the population comprising of elderly patients with neurocognitive disabilities.
Late presentation to hospital, high values of Thurston comorbidities and Alzheimer's disease, were correlated with poor outcomes. However, as a high proportion of patients were from elderly care institutions, an ascertainment bias with respect Alzheimer's disease seems likely.
Comments to authors:
- If possible, it would be useful to categorize outcome of patients treated before corticosteroid therapy was introduced, and after this became a standard of care.
2. As a high proportion of patients were from elderly care institutions, an ascertainment bias with respect Alzheimer's disease seems likely. This caveat should be included in the abstract and the discussion.
Author Response
Dear Reviewer,
Thank you for your analysis and the recommendation for improving the manuscript.
While a number of published studies in the last year described the epidemiological and clinical features of COVID-19 epidemic, we consider that our regional perspective in the beginning of pandemic could contribute to better understanding of peculiar dynamic of infection. On our knowledge, our department have managed the firs elderly people infected in long-term care facilities from Romania. This group of patients is consisting of over a half from the hospitalized cases in the first weeks of epidemic.
According up to date medical facts, our clinical and biological data allow the comparation of COVID-19 features and outcomes with other populations.
Answers to comments:
- The therapeutic considerations were excluded, as one of the reviewers suggested. We used the corticosteroids for severe and critical presentation, from the beginning, before to become the standard of care. Overall, 40% of patients received corticosteroids, but the correlation of surviving in critical or severe diseases was not significant for our patients.
- We have revised the abstract and the discussions with additional referenced regarding the influence of Alzheimer dementia on risk of death.
Kind regards,
Manuela Arbune
10 September 2021

Round 2
Reviewer 1 Report
Thank you for addressing the reviews made.